# Psychometric Properties of the Greek Version of the Connor–Davidson Resilience Scale (CD-RISC-10) in a Sample of Nurses

**DOI:** 10.3390/ijerph20186752

**Published:** 2023-09-13

**Authors:** Petros Galanis, Maria Elissavet Psomiadi, Chrysovalantis Karagkounis, Polyxeni Liamopoulou, Georgios Manomenidis, Georgios Panayiotou, Thalia Bellali

**Affiliations:** 1Clinical Epidemiology Laboratory, Faculty of Nursing, National and Kapodistrian University of Athens, 11527 Athens, Greece; 2Directorate of Operational Preparedness for Public Health Emergencies, Greek Ministry of Health, 10433 Athens, Greece; 3Department of Social Welfare, Education and Equality, Municipality of Katerini, 60133 Katerini, Greece; 4Department of Nursing, School of Health Sciences, International Hellenic University, 57001 Thessaloniki, Greece; 5Nursing Department (Didimoteicho Branch), International Hellenic University, 57400 Thessaloniki, Greece; 6Laboratory of Exercise, Health and Human Performance, Applied Sport Science Postgraduate Program, Department of Life Sciences, School of Sciences, European University Cyprus, Nicosia P.O. Box 22006, Cyprus; g.panayiotou@euc.ac.cy; 7Department of Health Sciences, School of Sciences, European University Cyprus, 6, Diogenes Str. Engomi, CY-2404, Nicosia P.O. Box 22006, Cyprus

**Keywords:** CD-RISC, translation, Greek, validation, resilience, nurses

## Abstract

Resilience has been defined as one’s ability to maintain a mental health state and overall well-being when undergoing grave stress or facing significant adversity. Numerous resilience-investigating research tools have been developed over the years, with the Connor–Davidson Resilience Scale (CD-RISC), a self-rated tool presenting valuable psychometric properties, remaining one of the most prominent. We aimed to translate and validate the brief CD-RISC-10 in a convenience sample of 584 nurses in Greece’s secondary and tertiary health care system. We conducted a confirmatory factor analysis and known-groups validity test and estimated the reliability of the CD-RISC-10. Our confirmatory factor analysis revealed that the scale had a unifactorial structure since all the model fit indices were very good. Moreover, the reliability of the CD-RISC-10 was very good since the acquired Cronbach’s alpha and McDonald’s omega were 0.924 and 0.925, respectively. Therefore, the Greek version of the CD-RISC-10 confirmed the factor structure of the original scale and had very good validity and reliability.

## 1. Introduction

Resilience is a psychological construct developed in the last quarter of the 20th century to simultaneously conceptualize the innate mechanisms and ‘individual’ characteristics thought to protect one from psychotic disorders [1,2,3,4,5]. It has been defined as a dynamic capacity fostered throughout one’s life course [2,4] via distinct mechanisms that are subject to numerous determinant factors [2,4,5]. Despite being primarily identified as a positive personality trait among children and psychiatric patients [6,7], nowadays, there is a consensus on the construct of resilience, namely, a metatheory depicting and rationalizing the continuous evolution of one’s innate predisposition to adapt to and recover from adversities rather quickly, which can be either enhanced or diminished under each respective life circumstance [8,9,10,11].

In particular, this construct’s origins date back to the early psychiatric literature, which primarily described it as a personality trait with a preventive effect on children under adverse circumstances that lessened their vulnerability [6]. Later on, resilience was recognized as an integral yet regular developmental trait with an interactive nature, and the term was assigned a dynamic character [10]. With the prominence of positive psychology and the consequence of kinship with respect to strengthening competencies instead of targeting flaws and limitations [12], resilience was acknowledged as the determining factor of one’s well-being, regardless of whether a person suffers from a psychiatric disease [12].

Even though there is no concrete definition of resilience, it has been recognized as a multitude of behaviours, thoughts, and actions [13,14] that enables individuals to retain a sense of coherence under distress, consequently causing them to adopt and manifest efficient coping skills to maintain their composure under challenging circumstances they perceive as manageable yet meaningful [15,16,17,18]. Pan-human contextual qualities have also been ascribed to the term, and it has thus been regarded as an ability related to self-regulation and the adoption of socially appropriate demeanours within the socio-cultural construct of communities [13,19].

Over the last four decades, research in the field has highlighted that resilience constitutes a multifaceted quality that, due to its dynamic core, varies significantly over time depending on the concurrent circumstances, one’s age, gender, ethnicity, and cultural background [7,20,21,22,23]. Resilience has been found to correlate positively with physical and mental well-being, self-efficacy, gratitude, and optimism and negatively with depression, post-traumatic stress disorder (PTSD), and generalized anxiety disorder [24,25,26,27,28,29].

A theory developed to further elaborate on this ever-changing nature of resilience is that developed by Richardson and colleagues [7], which suggests that one’s resilience capacity at a given time is the sole outcome of one’s “biopsychospiritual homeostasis”-processes-related ability to fully adapt one’s body, spirit, and mind in the circumstances one is under at a given point, despite any stressors, given their determined coping abilities, (whether constructive or not). This outcome, which might be positive or negative, can be manifested as a higher homeostasis level, i.e., an opportunity for personal growth, a return to one’s baseline homeostasis levels, a lower homeostasis level, or a problematic state replete with dysfunctional coping mechanisms and self-destructive behaviours [7].

It has been suggested that resilient persons seem more adaptive, especially within social settings that require an essential sense of morale [30], as they appear to be experiencing more positive emotions despite the adverse circumstances and challenges they may face [29]. This is why personality facets such as hope have been found to enhance one’s sense of self-efficacy, thereby provoking individuals to deploy more efficient and effective adaptive behaviours [7]. Furthermore, mutual support within groups has been positively associated with one’s mental health [31], thus reinforcing coping abilities [31]. Therefore, resilience has been recognized as a mental health index measuring one’s coping capacity. Besides recognizing hardiness, a protective factor against developing trauma and PTSD, resilience has been critical for health promotion and well-being in recent years [23].

In the early 1990s, Wagnild and Young developed the 25-item Resilience Scale [30] to explore resilience in adolescent populations. It is a research tool that has been deemed suitable for adult resilience assessment via its deployment in numerous studies [32,33,34,35,36,37,38]. This 25-item scale depicts resilience’s five primary components, namely, equanimity, perseverance, self-reliance, meaningfulness, and existential aloneness [32], while an abridged 14-item scale, with significant psychometric properties that have been translated into numerous languages [12,36], has been formulated based on the original scale to encompass stress-related protective factors that result in better coping outcomes [32]. From the validation processes carried out across different populations, it was suggested that scoring in the Resilience Scale was positively correlated with age due to the construct’s dynamic character, whereas no statistically significant correlation with gender was identified [12]. Many relevant scales have been established, with the Resilience Attitudes and Skills Profile, the Resilience Scale for Adults, and the Brief Resilience Scale [34,35,36,37] numbering among them.

However, one of the most prominent scales remains the Connor–Davidson Resilience Scale [39], a brief, 25-item, self-rated resilience tool with valuable psychometric properties [39]. The CD-RISC consists of 25 affirmations answered via a 5-item Likert scale, where “0” corresponds to “not true at all” and “4” corresponds to “true nearly all of the time”. The basis for rating lies in the frequency with which one experiences what the affirmations describe; hence, the total score ranges from 0 to 100, with the highest scores reflecting greater resilience [39]. Its preliminary analyses on differing samples from the general population, psychiatric inpatients, primary healthcare, and clinical trials established reference values for resilience or non-resilience. In particular, for the general population, an exploratory factor analysis highlighted the constructs of “personal competence, high standards and tenacity”; “trust in one’s instincts, tolerance of negative affect, and strengthening effect of stress”; “positive acceptance of change and secure relationships”; “control”; and “spirituality” [39].

In a recent study conducted on 744 critical care nurses, researchers intended to establish the factor structure of the CD-RISC abridged version while further validating it by analysing its construct validity [40]. The factors identified were “personal competence”, “perseverance”, and “leadership”, all portraying successful examples of coping mechanisms adopted by nurses to overcome the stressors of the Intensive Care Unit (ICU) environment, with eigenvalues > 1.0 and an explained variance of 59%. In addition, in this study, it was found that CD-RISC can differentiate between nurses that cope well after having experienced a traumatic event and those who do not, for which it was noted that those nurses that met the PTSD diagnostic criteria presented as less resilient [40].

Additionally, in a study assessing participants’ levels of resilience while identifying associated personal factors and physical activity behaviours, an interrelation between resilience and physical activity emerged [41]. Through the deployment of the CD-RISC in a sample of 93 New Zealand ICU nurses, resilience was found to be correlated positively with demanding factors related to physical jobs, including occupational physical demands, physical activity levels, moderate to vigorous occupational physical activity engagement, and dynamic standing at work, highlighting that resilient nurses present greater tolerance to high physical work stress [42].

The 25-item CD-RISC was applied in a 2020 study exploring social support, resilience, and mental health associations among 1521 Chinese healthcare professionals following the outbreak of the COVID-19 pandemic [43]. Personnel with less overall experience or less crisis management experience appeared less resilient, presenting significantly lower scores on the scale compared to the more experienced personnel with respect to tenacity, strength, and optimism. Resilience, especially with respect to tenacity and strength, was also a predictive factor for novel personnel’s mental health statuses amid adversity [44]. A corresponding study on 114 healthcare professionals sent to the Hubei area to support the regional health system’s operation identified occupation, education, and mental health training as resilience-determining factors [45]. Doctors, bachelor’s degree holders, and those who spent extended periods engaging in mental health training appeared more resilient, while resilience positively correlated with active coping and training [45]. In another study analysing 52 critical care professionals with the 10-item CD-RISC [44], resilience was found to be positively correlated with mental health, while a mediating role was ascribed to two burnout dimensions, “emotional exhaustion” and “depersonalization”, as well as “personal accomplishment”. A mixed-methods approach was adopted to estimate healthcare professionals’ distress and resilience statuses during the COVID-19 pandemic to study crucial resilience metatheory parameters [46].

In short, the CD-RISC-25 has been widely used in a great number of studies, presenting good psychometric properties in different settings and countries. However, there is significant disagreement regarding the factor structure of CD-RISC-25. For instance, Wu et al. found that the tool had four-factor structure when applied to a sample of employees [47], while Yu and Zhang found that it had a three-factor structure [48]. Since the factor structure of CD-RISC-25 has been unstable, Campbell-Sills and Stein refined the CD-RISC-25 and established a shorter form, i.e., the CD-RISC-10, with ten items [49]. 

Afterwards, several studies examined the psychometric properties of the CD-RISC-10 in different populations, settings, and cultures. In particular, Tu et al. confirmed the single-factor model of the CD-RISC-10 in a Chinese military personnel sample [50]. Also, they found that the CD-RISC-10 had good concurrent and predictive validity using an anxiety and depression scale [50]. Several other studies confirmed the unidimensional factor structure of the CD-RISC-10 [50,51,52,53,54,55,56,57]. Furthermore, the single-factor model of the CD-RISC-10 is strongly supported since it has been found in studies on different countries (e.g., the USA, the UK, China, France, Spain, Iran, and Nigeria) with different study populations (e.g., healthy individuals, chronic patients, young adults, and older adults). Additionally, a considerable number of studies have suggested that CD-RISC-10 has good convergent criteria and divergent validity when using a variety of tools measuring social support, self-efficacy, self-esteem, mastery, mental health, depression, and anxiety [50,51,53,54,55,56,57]. For instance, Neyer et al. found a strong positive correlation (r = 0.78) between CD-RISC-10 and another tool that measures resilience, namely, the 14-item Resilience Scale [55]. Moreover, several studies have shown the excellent internal consistency and reliability of the CD-RISC-10 by calculating Cronbach’s alpha coefficient and performing test–retest reliability analysis [52,53,54,55,56,57]. In particular, the corresponding Cronbach’s alpha coefficient ranged from 0.880 to 0.930. Also, the intraclass correlation coefficients in the test–retest reliability analysis ranged from 0.800 to 0.900 [51,52,53,54,55,56,57]. In conclusion, the CD-RISC-10 has superior psychometric properties, and some scholars have suggested replacing the CD-RISC-25 with the CD-RISC-10 to measure resilience [49,51,53].

A recent study examined the psychometric properties of the Greek version of the CD-RISC-25 in a sample of 244 healthy participants and 303 psychiatric patients [58]. However, until now, the Greek version of the CD-RISC-10 has not been validated. Moreover, the literature suggests that resilience among nurses is a significant internal psychological factor that promotes their ability to cope with difficult situations [59,60,61,62]. Thus, the validation of the CD-RISC-10 using a sample of nurses can give scholars a brief, simple, and valid tool with which to measure nurses’ resilience. Therefore, we aimed to translate the CD-RISC-10 into Greek, validate its psychometric properties, and investigate the resilience levels among a convenience sample of nurses currently working in the Greek public healthcare system. As mentioned above, the literature suggests a single-factor model of the CD-RISC-10. In this context, we expected also a single-factor model of the Greek version of the CD-RISC-10. For this reason, we performed confirmatory factor analysis to establish the structure of the Greek version of the CD-RISC10. Moreover, we hypothesized that different levels of resilience could arise according to several demographic and job characteristics, namely, gender, age, living arrangement, children, job position, educational level, and clinical experience. Finally, we examined the measurement invariance of the CD-RISC-10 since evidence on this issue is limited. 

## 2. Materials and Methods

### 2.1. Sample and Data Collection

A convenience sample of 584 male and female nurses (response rate 89.3%), recruited from 10 Greek public general hospitals from the secondary and tertiary health care systems, participated in this cross-sectional study from September to December 2022. The study’s description was uploaded to the hospitals’ official web page with the invitation to participate, while printed wall announcements were also placed in all hospital departments. After briefing the participants on the research procedures and the voluntary and anonymous nature of their participation throughout all the study’s stages, all subjects gave their informed consent for inclusion before participating. This study was conducted in accordance with the Declaration of Helsinki, and the protocol was approved by the Ethics Committee of the Faculty of Nursing of the National and Kapodistrian University of Athens (reference number; 417, September 2022).

### 2.2. Measures

#### The Connor–Davidson Resilience Scale (CD-RISC-10)

The Greek CD-RISC version consists of 10 affirmations describing different aspects of resilience corresponding to flexibility, self-efficacy, emotion regulation, optimism, and cognitive focus/maintaining attention under stress. Each affirmation is assigned to a 5-item Likert scale ranging from 0 (“not true”) to 4 (“true nearly all time”). Overall score can range from 0 to 40, with higher scores indicating greater resilience. The respondent is asked to provide his/her answer based on his/her experiences over the previous 30 days and whether those affirmations are true for him/her and to what extent.

The license to adapt the CD-RISC was granted by its developers following dedicated electronic communication. Translation into Greek was performed following the proposed procedure for intercultural adaptation of self-reference questionnaires [63]. Two independent, experienced researchers fluent in English translated the questionnaire into Greek. Consequently, the two texts were compared question by question until a consensus was reached. The revised text of the resulting Greek version was translated into English by two bilingual researchers who compared their translations for inconsistencies. The final text was compared with the original English scale’s text to confirm its linguistic accuracy. The cognitive debriefing process involved pre-testing the translated questionnaire on a few nurses to identify linguistic issues and adequate translation alternatives while assessing participants’ comprehension of the questions included. Due to COVID-19 restrictions, an electronic questionnaire with a completion time of approximately 10–15 min was constructed using Google Forms 

### 2.3. Data Analysis

Mean, standard deviation (SD), median, and minimum and maximum values are used to describe continuous variables, and numbers (percentages) are used to describe categorical variables. Moreover, we calculated skewness and kurtosis values for the ten items of CD-RISC-10. The skewness and kurtosis values between −1.00 and 1.00 are indicative of normal distribution.

We conducted confirmatory factor analysis (CFA) to confirm the unifactorial structure of the Greek version of CD-RISC-10. The ten items of CD-RISC-10 followed a normal distribution; thus, we used a maximum likelihood estimator. We employed a Maximum Likelihood estimator with 200 bootstrap samples. We checked the goodness of fit indices in CFA by measuring the following: chi-square/degree of freedom (x^2^/df); root mean square error of approximation (RMSEA); goodness of fit index (GFI); adjusted goodness of fit index (AGFI); Tucker–Lewis index (TLI); incremental fit index (IFI); normed fit index (NFI); comparative fit index (CFI). The acceptable value for x^2^/df is less than 5, that for RMSEA is less than 0.10, and that for all other indices is higher than 0.90 [64,65,66,67]. Additionally, we calculated standardized regression weights between the 10 items and the one factor. Moreover, we examined configural and metric invariance between the two gender groups. In case of configural invariance, we examined the goodness of fit indices in the CFA mentioned above. In case of metric invariance, we calculated *p*-values, with values higher than 0.05 supporting metric invariance [68]. We used AMOS version 21 (Amos Development Corporation, Wexford, PA, USA, 2018) to conduct CFA.

Additionally, we estimated the known-groups validity of the CD-RISC-10. In this case, we assessed the relationship between nurses’ demographic and job characteristics and the total score on CD-RISC-10. We used Pearson’s correlation coefficient to assess the correlation between age and CD-RISC-10 score, Spearman’s correlation coefficient to assess the correlation between experience and CD-RISC-10 score, and independent samples *t*-test to assess the relationship between gender, marital status, children, job position, and educational level and CD-RISC-10 score. After bivariate analysis, we performed multivariable linear regression analysis to eliminate confounding and control for other variables. We included all demographic and job characteristics as independent variables in a final multivariable linear regression model to estimate the independent effect of each variable. In this case, we present adjusted coefficient betas, 95% confidence intervals (CIs), and *p*-values. *p*-values less than 0.05 were considered statistically significant. We used IBM SPSS 21.0 (IBM Corp. Released 2012. IBM SPSS Statistics for Windows, Version 21.0. Armonk, NY, USA: IBM Corp.) to perform the analysis.

Finally, we estimated the reliability of the CD-RISC-10 by calculating the following: Cronbach’s alpha, McDonald’s Omega, corrected item–total correlations, and Cronbach’s alpha when a single item was deleted for the ten items of the CD-RISC-10. Cronbach’s alpha and McDonald’s Omega values higher than 0.7 are considered acceptable [69].

## 3. Results

The respondents included 584 registered nurses. The mean age of the respondents was 45.8 years (SD: 6.4), ranging from 23 to 65. In our sample, 51.6% of the respondents were females and 48.4% were males. Most nurses lived with their family/partner/spouse (86.6%) and had children (75.0%). Most nurses worked in internal departments (76.5%). One out of two nurses possessed an MSc/PhD diploma (52.5%). The mean work experience (in years) was 14.1 (SD: 9.8), ranging from 1 to 31. The demographic and job characteristics of the sample are shown in Table 1.

Descriptive statistics for the CD-RISC-10 are shown in Table 2. The mean score for the CD-RISC-10 was 22.6 (SD: 8.0). The minimum and maximum scores were 1 and 40, respectively. The skewness values for the ten items ranged from −0.44 to 0.06, while the kurtosis values ranged from −1.00 to 0.07. Therefore, the ten items followed a normal distribution since the skewness and kurtosis values were between −1.00 and 1.00.

The Cronbach’s alpha value for the CD-RISC-10 was 0.924, and the McDonald’s omega value was 0.925, indicating excellent reliability. Similarly, Cronbach’s alpha for the scale decreased if any item was deleted (Table 2). Additionally, the corrected item–total correlation coefficients ranged from 0.636 to 0.776 (*p*-value < 0.001 in all items), indicating very good reliability (Table 2).

The results of the CFA for the CD-RISC-10 are shown in Figure 1. The fit indices of the model were very good: x^2^/df = 4.662, RMSEA = 0.079, GFI = 0.951, AGFI = 0.905, TLI = 0.955, IFI = 0.972, NFI = 0.965, and CFI = 0.972. Moreover, the standardized regression weights between the 10 items and the one factor ranged from 0.670 to 0.810 (*p* < 0.001 in all cases). Therefore, the Greek version of the CD-RISC-10 confirmed the factor structure of the original scale.

Then, we examined invariance with respect to gender (males vs. females). By testing configural invariance, we found that the single-factor CFA model was a very good fit for males and females. In particular, we obtained the following results: x^2^/df = 2.759, RMSEA = 0.055, GFI = 0.947, AGFI = 0.901, TLI = 0.957, IFI = 0.974, NFI = 0.959, and CFI = 0.973. Moreover, we found that metric invariance had been established for gender since the *p*-value was 0.377.

The known-groups validity of the CD-RISC-10 is shown in Table 3. We found that nurses with children had higher levels of resilience (*p* < 0.001). In particular, the mean CD-RISC-10 score for nurses with children was 23.6, while it was 20.6 for those without children. Also, we found that a higher educational level was associated with resilience since the mean CD-RISC-10 score for nurses with a PhD/MSc was 23.8, while it was 22.22 for those without a PhD/MSc (*p* = 0.02). Females’ resilience was higher than males (23.2 vs. 22.1), but this difference was not statistically significant (*p* = 0.11). Then, we performed multivariable linear regression analysis to eliminate confounding. The final multivariable model confirmed the results from the bivariate analysis. In particular, we found that nurses with children (adjusted coefficient beta = 4.06, 95% CI = 2.46 to 5.67, *p* < 0.001) and those with a PhD/MSc diploma (adjusted coefficient beta = 2.75, 95% CI = 1.44 to 4.05, *p* < 0.001) had higher resilience (Table 4).

## 4. Discussion

Our aim was to translate and validate the CD-RISC-10 using a sample of nurses in Greece. Since the literature strongly suggests a single-factor model of the CD-RISC-10, we performed confirmatory factor analysis to investigate the structure of the tool in our sample. The results of our CFA are consistent with the findings of other studies confirming the single-factor model of the tool. Numerous studies support our finding that the CD-RISC-10 is a unidimensional tool for measuring resilience [50,51,52,53,54,55,56,57]. Our study investigated, for the first time, the validity of the CD-RISC-10 using a sample of nurses. Previous studies investigated the validity of this tool in different study populations, such as patients, healthy individuals, and younger and older adults. Moreover, these studies were conducted in different regions, including Europe (the United Kingdom, France, and Spain), North America (the USA), Asia (China), and Africa (Nigeria). A similar study was performed recently in Greece, but the corresponding researchers assessed the validity of the CD-RISC-25 and not that of the CD-RISC-10 with respect to healthy individuals and psychiatric patients [58].

Additionally, we found that the CD-RISC-10 had great internal consistency in our sample, presenting Cronbach’s alpha and McDonald’s omega values greater than 0.92. The literature shows that the reliability of the CD-RISC-10 is excellent since the obtained Cronbach’s alpha values ranged from 0.880 to 0.930 in studies on different countries [52,53,54,55,56,57]. Also, investigators that performed test–retest reliability analysis found very high intraclass correlation coefficients (greater than 0.800), thus supporting the high reliability of the CD-RISC-10.

Moreover, our multivariable analysis identified that levels of resilience were higher among nurses with higher educational levels. In particular, nurses with a PhD/MSc diploma had higher score on the CD-RISC-10 than those without a PhD/MSc diploma. This finding is consistent with the results of a recent study where scholars found a positive correlation between resilience and educational level in a sample of intensive care unit nurses across the United States [45]. In particular, 50% of registered nurses showed high levels of resilience, while the respective percentage for nursing assistants was 23%. Moreover, a study incorporating 1012 nurses in Greece found that resilient nurses were better educated after the elimination of several confounders [69]. Higher education offers nurses higher critical reflective abilities and autonomy [70]. Additionally, highly educated nurses with better theoretical knowledge can more easily develop and implement well-planned strategies and positive reappraisals in a clinical setting [71]. In this context, a high educational level can improve nurses’ resilience in order to deal with difficulties in clinical environments.

Furthermore, after eliminating confounders, we found that nurses with children presented higher levels of resilience compared to those without children. This finding was also observed in the study by Afshari and colleagues [72]. One possible explanation for this finding could be that having children and worrying about oneself and one’s children becoming infected with COVID-19 are among the factors contributing to anxiety and stress among nurses, which may play an important role in reducing nurses’ resilience.

Concerning gender, in our sample, female nurses appeared more resilient than males, constituting a relationship that was not statistically significant. Similar findings over the years have emphasized that despite the differences in scoring in the CD-RISC scales between males and females, gender does not affect one’s levels of resilience, as no statistically significant relations have been identified in corresponding studies (i.e., focusing on adult nurses and adult healthcare professionals) [39,40,41,42,43,44]. Interestingly, a study concerning the challenges and adversities faced by healthcare professionals when combating the COVID-19 pandemic concluded that neither gender nor age impact resilience capacity [73]. Our findings agree with this, as no statistically significant association was found between participants’ age and scores on the scale.

Our study had several limitations. The convenience-sampling method adopted for this survey might entail potential selection bias. However, our study sample exhibited a gender distribution and demographics analogous to those recorded for the Greek population in general [74], thus slightly, if at all, affecting the tool’s psychometric evaluation, given women’s predominant role in the nursing profession [75]. Additionally, the source population’s demographic and professional characteristics with respect to the hospitals participating in the study remain unknown, thereby impeding our ability to compare the study sample with the source population. A significant strength of the study is that it offers a reliable tool whose use in future research could provide a deeper understanding of the factors that determine resilience, especially in heavily burdened professional groups such as nursing personnel.

## 5. Conclusions

The Greek version of the CD-RISC-10 confirmed the factor structure of the original scale and validated its prominent psychometric properties. The Greek version of the CD-RISC-10 showed great validity and reliability, constituting a highly suitable tool for investigating the Greek population’s resilience capacity.

The importance of investigating healthcare professionals’ resilience with a scale such as the CD-RISC lies with the scale’s main properties, which, apart from remaining unchanged independently of the scale’s length and cultural adaptations, facilitate the identification of factors that may enhance or hinder resilience, all the while indicating adequate strategies for improving one’s coping skills [76]. Investigating healthcare professionals’ resilience through the CD-RISC in different populations while investigating additional contributing parameters would assist in ascertaining any potentially contributing factors with respect to burnout and adverse outcomes, creating a series of implications leading to a change in the organizational culture of healthcare organizations and even encouraging the adoption of human-resource-management-resilience-dedicated policies [63].

Nowadays, healthcare professionals significantly struggle due to the impacts of many stressors apart from their personal lives and their professions. Occupational stressors, including time constraints, scheduling, burdened workloads, spiritual and ethical distress, uncertainty, and a sense of cancellation, trigger negative emotional and behavioural responses to the suffering of patients these professionals care for [42]. Healthcare professionals, especially nurses in closed departments, are predominantly exposed to grave stressors, as indicated by the high prevalence of burnout syndrome [40], a secondary indicator of their challenged mental health and, consequently, reduced resilience capacity. Given this predicament, additional studies in the field are required to explore this population’s levels of resilience and provide a better understanding of the factors that primarily affect them.

Future studies should involve other healthcare professionals as well and focus on obtaining a robust tool suitable for assessing levels of resilience among professionals in the field and exploring the impacts biological parameters and behavioural interventions, such as sleep patterns, nutritional habits, physical activity/exercise participation, etc., might pose.

## Figures and Tables

**Figure 1 ijerph-20-06752-f001:**
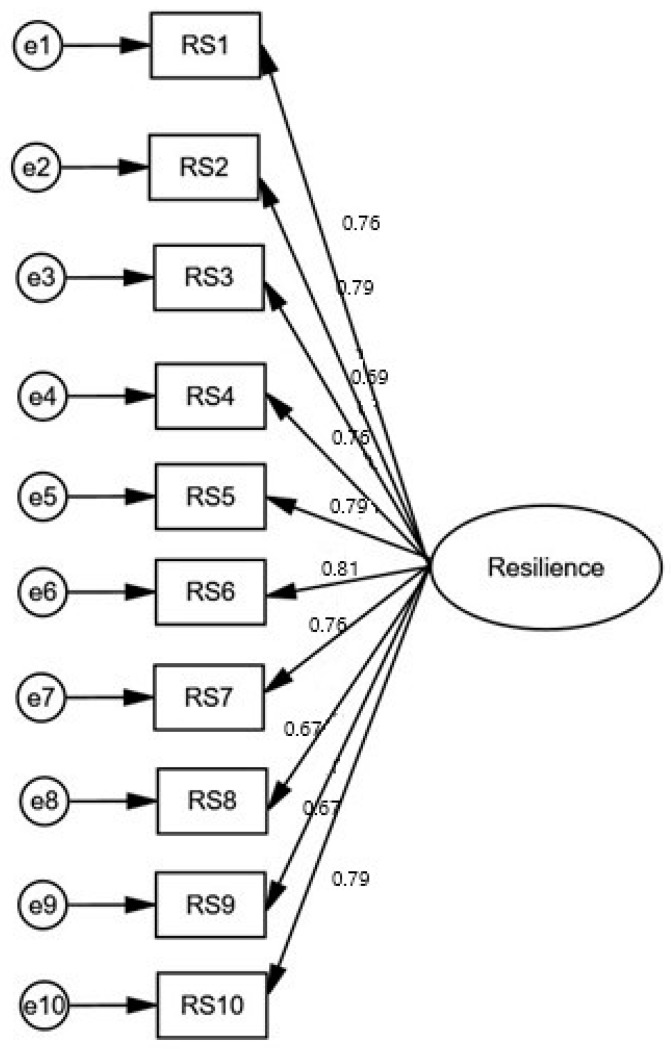
Confirmatory factor analysis for the CD-RISC-10.

**Table 1 ijerph-20-06752-t001:** Demographic and job characteristics of the sample.

*Characteristics*	*n*	%
*Gender (n = 583)*		
Males	282	48.4
Females	301	51.6
*Age (n = 577)*	45.8 ^a^	6.4 ^b^
*Living arrangement (n = 583)*		
Alone	78	13.4
With family/partner/spouse	506	86.6
*Children (n = 565)*		
No	141	25.0
Yes	424	75.0
*Job Position (n = 584)*		
Internal departments	447	76.5
ICU	137	22.5
*MSc/PhD (n = 584)*		
No	277	47.5
Yes	307	52.5
*Clinical experience (n = 577)*	14.1 ^a^	9.8 ^b^

^a^ mean; ^b^ standard deviation.

**Table 2 ijerph-20-06752-t002:** Descriptive statistics for the CD-RISC-10 (N = 584).

CD-RISC-10 Items	Mean	Standard Deviation	Median	Skewness	Kurtosis	Factor Weights	Cronbach’s Alpha When a Single Item Was Deleted	Corrected Item–Total Correlation Coefficient
Adapt to change	2.57	0.98	3	−0.33	−0.40	0.76	0.915	0.728
Deal with whatever comes my way	2.29	1.00	2	−0.14	−0.70	0.79	0.914	0.755
See humorous side of things	1.79	1.24	2	0.03	−1.00	0.66	0.922	0.636
Stress makes me stronger	1.99	1.19	2	−0.15	−0.89	0.76	0.915	0.734
Bounce back after illness or injury	2.25	1.15	2	−0.21	−0.82	0.79	0.913	0.759
Believe I can achieve goals despite obstacles	2.44	0.97	2	−0.17	−0.35	0.81	0.913	0.776
Under pressure, I stay focused	2.33	0.92	2	−0.09	−0.19	0.76	0.916	0.721
Not easily discouraged by failure	2.06	0.99	2	0.06	−0.35	0.67	0.920	0.637
Think of myself as a strong person when facing challenges	2.68	0.88	3	−0.44	0.07	0.67	0.920	0.637
Able to handle unpleasant feelings	2.24	0.97	2	−0.04	−0.61	0.79	0.913	0.758

**Table 3 ijerph-20-06752-t003:** Known-groups validity of the CD-RISC-10.

Characteristics	Mean CD-RISC-10 Score	Standard Deviation	*p*-Value
*Gender (n = 583)*			0.11 ^a^
Males	22.1	7.9	
Females	23.2	8.1	
*Age (n = 577)*		0.03 ^b^	0.50 ^b^
*Living arrangement (n = 583)*			0.44 ^a^
Alone	24.0	8.1	
With family/partner/spouse	23.0	7.8	
*Children (n = 565)*			<0.001 ^a^
No	20.6	7.4	
Yes	23.6	8.0	
*Job position (n = 584)*			0.46
Internal departments	22.5	7.9	
ICU	23.1	8.2	
*PhD/MSc (n = 584)*			0.02 ^a^
No	22.2	8.2	
Yes	23.8	7.5	
*Clinical experience (n = 577)*		0.03 ^c^	0.50 ^c^

^a^ independent samples *t*-test, ^b^ Pearson’s correlation coefficient, ^c^ Spearman’s correlation coefficient.

**Table 4 ijerph-20-06752-t004:** Multivariable linear regression analysis, with score on the CD-RISC-10 scale serving as the dependent variable.

Independent Variables	Adjusted Coefficient Beta	95% CI for Beta	*p*-Value
Females vs. males	1.02	−0.27 to 2.31	0.121
Age	−0.37	−1.55 to 0.80	0.533
Living alone vs. living with others	1.23	−0.15 to 1.89	0.102
Children	4.06	2.46 to 5.67	<0.001
ICU vs. internal departments	0.86	0.65 to 2.37	0.263
PhD/MSc	2.75	1.44 to 4.05	<0.001
Clinical experience	0.04	−0.07 to 0.15	0.446

CI: confidence interval.

## Data Availability

The data presented in this study are available on request from the corresponding author. The data are not publicly available.

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
