# Peer review of "Psychometric Properties of the Greek Version of the Connor–Davidson Resilience Scale (CD-RISC-10) in a Sample of Nurses"

_ijerph, 2023, doi:10.3390/ijerph20186752_

Round 1

Reviewer 1 Report

The study is aimed at evaluating the psychometric properties of the Greek Version of the Connor- Davidson Resilience Scale (CD-RISC-10) in a sample of nurses in Greece. The manuscript is well-written and well-articulated. Below are a few minor comments:

·       Introduction, lines 33-35: Is resilience only meant to protect individuals from psychotic disorders? If it is an old conceptualization, it should be stated more clearly.

·       Line 66: Richardson et al. lacks the reference number (it is quoted only later).

·       Line 141: is “personal accomplishment” a “burnout dimension”?

·       Line 252: spouse, NOT spouce

·       As a main finding, the authors mention the association between educational level and resilience. However, they do not discuss this interesting result in the Discussion section.

·       Lines 339-340: I would encourage the authors to say a bit more about the “adequate strategies for improving one’s coping skills”, also providing some examples. 

The English is fine, I just detected a typo (see comments above).

Author Response

Dear Reviewer,

Thank you very much for the peer review of the manuscript “Psychometric properties of the Greek Version of the Connor- Davidson Resilience Scale (CD-RISC-10) in a sample of nurses”. Thank you for your comments, which have improved the quality of the manuscript.

We have addressed all the comments (highlighted in yellow) in the revised text. Also, we made changes in the manuscript according to the other Reviewers’ instructions.

Please, find below an item-by-item answer to your comments. Hoping the revised manuscript fulfils the journal’s standards, we thank you for your courtesy.

We are looking forward to your response.

Best Regards

The authors

The study is aimed at evaluating the psychometric properties of the Greek Version of the Connor- Davidson Resilience Scale (CD-RISC-10) in a sample of nurses in Greece. The manuscript is well-written and well-articulated. Below are a few minor comments:

  • Introduction, lines 33-35: Is resilience only meant to protect individuals from psychotic disorders? If it is an old conceptualization, it should be stated more clearly.

Answer: Done

For articulating more properly that originally the concept of resilience was developed to describe protective factors against psychotic disorders we rephrased the passage as “ Resilience comprises a psychological construct developed in the last quarter of the 20th century to conceptualize at the time the innate mechanisms and 'individuals’ characteristics considered protective from psychotic disorders [1-5].”

  • Line 66: Richardson et al. lacks the reference number (it is quoted only later).

Answer: Done

The reference number has been placed more adequately.  The text currently is in the form “…Richardson and colleagues[7]..”

  • Line 141: is “personal accomplishment” a “burnout dimension”?

Answer: Done

Following your comment the passage has been restated as “…resilience was found to be positively correlated to mental health, all the while casting a mediating role to two burnout dimensions, “emotional exhaustion” and“depersonalization”, as well as “personal accomplishment”.

  • Line 252: spouse, NOT spouce

Answer: Done

  • As a main finding, the authors mention the association between educational level and resilience. However, they do not discuss this interesting result in the Discussion section.

 Answer: Done

We added the following text in the Discussion section.

Moreover, our multivariable analysis identified that levels of resilience was higher among nurses with higher educational level. In particular, nurses with a PhD/MSc diploma had higher score on CD-RISC-10 than those without a PhD/MSc diploma. This finding is consistent with a recent study where scholars found a positive correlation between resilience and educational level in a sample of intensive care units nurses across the United States [45]. In particular, 50% of registered nurses showed high levels of resilience, while the respective percentage for nursing assistants was 23%. Moreover, a study with 1012 nurses in Greece found that resilient nurses were better educated after elimination of several confounders[69]. Higher education offers nurses higher critical reflective practice and autonomy[70]. Additionally, highly educated nurses with a better theoretical knowledge can more easily develop and implement planful problem-solving and positive reappraisal in clinical setting[71]. In this context, high educational level can improve nurses’ resilience to deal with difficulties in clinical environments.

  • Lines 339-340: I would encourage the authors to say a bit more about the “adequate strategies for improving one’s coping skills”, also providing some examples. 

Answer:

Dear Reviwer, the Discussion has been rewritten. Given its current form, elaborating more on those concepts was not deemed required.

Reviewer 2 Report

In this study, the authors have examined the psychometric properties of the Greek Version of the Connor-Davidson Resilience Scale (CD-RISC-10) in a sample of nurses.

Overall, the manuscript needs to be improved significantly in content and writing quality (The manuscript could benefit from a revision by a native English speaker). More specifically, the manuscript lacks a thorough literature review on the CD-RISC-10, which is needed for the Introduction and for discussing the results of the current study. Also, the authors need to clearly explain their data analysis plan and present their results more clearly. The discussion also lacks depth. I have outlined my recommendations for the authors in the following:

Introduction:
- Overall, the introductory section lacks depth and fails to provide a comprehensive analysis. For instance, there is no detailed explanation on whether the psychometric properties of CD-RISC-10 has been examined in other cultures and countries; what were the results? This is very important that authors provide a literature review of the CD-RISC-10 in the Introduction. In addition to all this, I recommend authors use the following article: Tsigkaropoulou, E., Douzenis, A., Tsitas, N., Ferentinos, P., Liappas, I., & Michopoulos, I. (2018). Greek version of the Connor-Davidson resilience scale: Psychometric properties in a sample of 546 subjects. in vivo, 32(6), 1629-1634. It is not entirely clear from the published study why a new translation was needed.

- Please also provide some hypotheses on what you expect about the factor structure and external correlates that you have used to examine the validity of the CD-RISC-10.

Method:
- Data Analysis section is too weak. Please provide more information.

Results:
-  Table 1 (Demographic and job characteristics of the sample) is difficult to follow and there are several discrepancies with the data reported in the text. Please revise Table 1 and come up with a more clear one.

- In Table 2 (Descriptive statistics for the CD-RISC-10), I feel it is less important to show the minimum and maximum values, it would be more useful to show, for example, the skewness and kurtosis. After all, they determine the method of factor analysis. The data analysis section incorrectly states: „The scale followed the normal distribution; thus, we used the maximum likelihood estimator”. Factor analysis was performed on the items, so their distributions are of interest, not the scale.

- Table 3 does not give much useful information after the factor weights (Figure 1), so I think the Corrected item-total correlation coefficient here could be included in Table 2, as well as the factor weights, which can only be read on the Figure 1 (Confirmatory factor analysis for the CD-RISC-10), in very small font.

- The validity analysis is also not very thorough. The following statement reflects this: „Known-groups validity of the CD-RISC-10 is shown in Table 4. We found that nurses with children had higher levels of resilience (p<0.001).It is not possible that age has an influential role here that would have been worth controlling for?

There are a lot of methodological inaccuracies in the descriptions, so they should be checked very carefully: e.g. „Females’ resilience was higher than males (23.2 vs. 22.1) but this relationship was not statistically significant (p=0.01).” This is not about correlation, and in addition, we see a significant p value.

A more in-depth psychometric analysis would be to estimate the number of dimensions for the factor structure: e.g., parallel analysis. Or a simple comparison of gender in addition to controlling for measurement invariance between the two gender groups.

Discussion:
- Discussion lacks depth and is too weak. The findings should be interpreted in the context of the results from previous studies. In the current format, the Discussion is mostly a repetition of the results section. What results did previous studies on the psychometric of the CD-RISK-10 yield? What about convergent validity? A literature review on the CD-RISK-10 is needed in the Introduction which could be used to discuss the results from the current study in the Discussion part.
- Please discuss your findings in the context of previous findings while also trying to avoid only reporting the results.

References:
- Please include DOI for the references.

It is crystal clear that this work needs much work in terms of literature review and interpretation of psychometric analysis. The number of references is very low. Please try to read and integrate more works

Regards

Author Response

Dear Reviewer,

Thank you very much for the peer review of the manuscript “Psychometric properties of the Greek Version of the Connor- Davidson Resilience Scale (CD-RISC-10) in a sample of nurses”. Thank you for your comments, which have improved the quality of the manuscript.

We have addressed all the comments (highlighted in yellow) in the revised text. Also, we made changes in the manuscript according to the other Reviewers’ instructions.

Please, find below an item-by-item answer to your comments. Hoping the revised manuscript fulfils the journal’s standards, we thank you for your courtesy.

We are looking forward to your response.

Best Regards

The authors

In this study, the authors have examined the psychometric properties of the Greek Version of the Connor-Davidson Resilience Scale (CD-RISC-10) in a sample of nurses.

Overall, the manuscript needs to be improved significantly in content and writing quality (The manuscript could benefit from a revision by a native English speaker). More specifically, the manuscript lacks a thorough literature review on the CD-RISC-10, which is needed for the Introduction and for discussing the results of the current study. Also, the authors need to clearly explain their data analysis plan and present their results more clearly. The discussion also lacks depth. I have outlined my recommendations for the authors in the following:

Introduction:
- Overall, the introductory section lacks depth and fails to provide a comprehensive analysis. For instance, there is no detailed explanation on whether the psychometric properties of CD-RISC-10 has been examined in other cultures and countries; what were the results? This is very important that authors provide a literature review of the CD-RISC-10 in the Introduction. In addition to all this, I recommend authors use the following article: Tsigkaropoulou, E., Douzenis, A., Tsitas, N., Ferentinos, P., Liappas, I., & Michopoulos, I. (2018). Greek version of the Connor-Davidson resilience scale: Psychometric properties in a sample of 546 subjects. in vivo, 32(6), 1629-1634. It is not entirely clear from the published study why a new translation was needed.

- Please also provide some hypotheses on what you expect about the factor structure and external correlates that you have used to examine the validity of the CD-RISC-10.

Answer: Done

We expanded the Introduction section providing studies that have examined the psychometric properties of CD-RISC-10 in different settings and populations. Moreover, we added the article that you mentioned explaining the reasons that a new translation was needed. Also, we presented our hypotheses in detail.

We add the following text in the introduction section:

In short, CD-RISC-25 has been widely used in a great amount of studies presenting good psychometric properties in different setting and countries. However, there is a significant disagreement regarding the factor structure of CD-RISC-25. For instance, Wu et al. found a 4-factor structure of the tool in a sample of employees[47], while Yu & Zhang found a 3-factor structure[48]. Since the factor structure of CD-RISC-25 has been unstable, Campbell-Sills & Stein refined CD-RISC-25 and established a shorter form, i.e. the CD-RISC-10 with ten items[49].

Afterwards, several studies examined the psychometric properties of CD-RISC-10 in different populations, settings, and cultures. In particular, Tu et al. confirmed the single-factor model of the CD-RISC-10 in a Chinese military personnel sample[50]. Also, they found that the CD-RISC-10 had a good concurrent and predictive validity using an anxiety and a depression scale. Several other studies confirmed the unidimensional factor structure of the CD-RISC-10[51-57]. Furthermore, the single-factor model of the CD-RISC-10 is strongly recommended since it is found in studies in different countries (e.g. USA, United Kingdom, China, France, Spain, Iran, Nigeria) with different study populations (e.g. healthy individuals, chronic patients, young adults, older adults). Additionally, a considerable amount of studies suggested that CD-RISC-10 has good convergent, criterion and divergent validity using a variety of tools measuring social support, self-efficacy, self-esteem, mastery, mental health, depression, and anxiety[50-51,53-57]. For instance, Neyer et al. found a strong positive correlation (r=0.78) between CD-RISC-10 and another tool that measures resilience, namely 14-item Resilience Scale[55]. Moreover, several studies showed the excellent internal consistency and reliability of CD-RISC-10 by calculating Cronbach’s alpha coefficient and performing test-retest reliability analysis. In particular, Cronbach’s alpha coefficient ranged from 0.880 to 0.93[52-57]. Also, intraclass correlation coefficients in test-retest reliability analysis ranged from 0.800 to 0.900. In conclusion, CD-RISC-10 has superior psychometric properties [51-57]and some scholars suggested the replacement of CD-RISC-25 with the CD-RISC-10 to measure resilience in studies[49,51,53].

A recent study examined the psychometric properties of Greek version of CD-RISC-25 in a sample of 244 healthy participants and 303 psychiatric patients[58]. However, until now the Greek version of CD-RISC-10 has not been validated. Moreover, literature suggests that resilience among nurses is a significant internal psychological factor that promotes their ability to cope with difficult situations[59-62]. Thus, validation of CD-RISC-10 in a sample of nurses can give scholars a brief, simple, and valid tool to measure nurses’ resilience. Therefore, we aimed to translate into Greek and validate the psychometric properties of the CD-RISC-10 and investigate the resilience levels among a convenience sample of nurses currently working in the Greek public healthcare system. As mentioned above, literature suggests a single-factor model of the CD-RISC-10. In this context, we expected also a single-factor model of the Greek version of CD-RISC-10. For that reason, we performed confirmatory factor analysis to establish the structure of the Greek version of CD-RISC10. Moreover, we hypothesized that different levels of resilience could be arisen according to several demographic and job characteristics, namely gender, age, living arrangement, children, job position, educational level, and clinical experience. Finally, we examined the measurement invariance of CD-RISC-10 since evidence on this issue is limited.

Method:
- Data Analysis section is too weak. Please provide more information.

Answer: Done

We expanded our analysis according to your comments. Thus, we added the following sentences in Data analysis section (Please, see also the manuscript):

Moreover, we calculated skewness and kurtosis values for the ten items of CD-RISC-10. Skewness and kurtosis values between -1.00 and 1.00 are indicative of normal distribution.

The ten items of CD-RISC-10 followed the normal distribution; thus, we used the maximum likelihood estimator. We employed the Maximum Likelihood estimator with 200 bootstrap samples.

Moreover, we examined configural and metric invariance between the two gender groups. In case of configural invariance, we examined the goodness of fit indices in CFA mentioned above. In case of metric invariance, we calculated p-value with values higher than 0.05 support metric invariance.

After bivariate analysis, we performed multivariable linear regression analysis to eliminate confounding and control for other variables. We included all demographic and job characteristics as independent variables in a final multivariable linear regression model to estimate the independent effect of each variable. In that case, we present adjusted coefficient betas, 95% confidence intervals (CIs), and p-values.

Results:
-  Table 1 (Demographic and job characteristics of the sample) is difficult to follow and there are several discrepancies with the data reported in the text. Please revise Table 1 and come up with a more clear one.

Answer: Done

Dear Reviewer, we apologize for discrepancies between text and Table 1. We re-wrote Table 1 and revised the text. Please, see the first paragraph in Results section and Table 1.

- In Table 2 (Descriptive statistics for the CD-RISC-10), I feel it is less important to show the minimum and maximum values, it would be more useful to show, for example, the skewness and kurtosis. After all, they determine the method of factor analysis. The data analysis section incorrectly states: „The scale followed the normal distribution; thus, we used the maximum likelihood estimator”. Factor analysis was performed on the items, so their distributions are of interest, not the scale.

Answer: Done

We replaced “The scale followed the normal distribution; thus, we used the maximum likelihood estimator” with “The ten items of CD-RISC-10 followed the normal distribution; thus, we used the maximum likelihood estimator.”

Also, we removed minimum and maximum values and added skewness and kurtosis values in Table 2. Moreover, we add the following text in Results section:

Skewness values for the ten items ranged from -0.44 to 0.06, while kurtosis values ranged from -1.00 to 0.07. Therefore, the ten items followed normal distribution since skewness and kurtosis values were between -1.00 and 1.00.

- Table 3 does not give much useful information after the factor weights (Figure 1), so I think the Corrected item-total correlation coefficient here could be included in Table 2, as well as the factor weights, which can only be read on the Figure 1 (Confirmatory factor analysis for the CD-RISC-10), in very small font.

Answer: Done

We included the Corrected item-total correlation coefficient in Table 2. Moreover, we added factor weights in Table 2, and we remove Table 3.

- The validity analysis is also not very thorough. The following statement reflects this: „Known-groups validity of the CD-RISC-10 is shown in Table 4. We found that nurses with children had higher levels of resilience (p<0.001).” It is not possible that age has an influential role here that would have been worth controlling for?

Answer: Done

We performed multivariable linear regression analysis to control for other variables. We add Table 4. We added the following text in the Results section:

 Females’ resilience was higher than males (23.2 vs. 22.1) but this difference was not statistically significant (p=0.11). Then, we performed multivariable linear regression analysis to eliminate confounding. The final multivariable model confirmed the results from bivariate analysis. In particular, we found that nurses with children (adjusted coefficient beta = 4.06, 95% CI = 2.46 to 5.67, p < 0.001) and those with PhD/MSc (adjusted coefficient beta = 2.75, 95% CI = 1.44 to 4.05, p < 0.001) had higher resilience (Table 4).

There are a lot of methodological inaccuracies in the descriptions, so they should be checked very carefully: e.g. „Females’ resilience was higher than males (23.2 vs. 22.1) but this relationship was not statistically significant (p=0.01).” This is not about correlation, and in addition, we see a significant p value.

Answer: Done

Thank you for your sharp eye. We replaced “Females’ resilience was higher than males (23.2 vs. 22.1) but this relationship was not statistically significant (p=0.01).” with “Females’ resilience was higher than males (23.2 vs. 22.1) but this difference was not statistically significant (p=0.11).”

A more in-depth psychometric analysis would be to estimate the number of dimensions for the factor structure: e.g., parallel analysis. Or a simple comparison of gender in addition to controlling for measurement invariance between the two gender groups.

Answer: Done

We examined the configural and metric invariance between the two gender groups. We added the following text in the results:

Then, we examined invariance with respect to gender (males vs. females). By testing configural invariance, we found that the 1-factor CFA model was a very good fit for males and females. In particular, we found that x2/df = 2.759, RMSEA = 0.055, GFI = 0.947, AGFI = 0.901, TLI = 0.957, IFI = 0.974, NFI = 0.959, and CFI = 0.973. Moreover, we found that metric invariance was established for gender since p-value was 0.377.

Discussion:
- Discussion lacks depth and is too weak. The findings should be interpreted in the context of the results from previous studies. In the current format, the Discussion is mostly a repetition of the results section. What results did previous studies on the psychometric of the CD-RISK-10 yield? What about convergent validity? A literature review on the CD-RISK-10 is needed in the Introduction which could be used to discuss the results from the current study in the Discussion part.

- Please discuss your findings in the context of previous findings while also trying to avoid only reporting the results.

Answer: Done

We rewrote the discussion according to your comment. We compared our factor structure with other models in the literature. Moreover, we expanded the Introduction section providing studies that have examined the psychometric properties of CD-RISC-10 in different settings and populations. Then, we used the studies in the Introduction section to discuss our results in the Discussion section. Additionally, we removed all results from the Discussion section.

We rewrote the Discussion section as following.

Our aim was to translate and validate the CD-RISC-10 in a sample of nurses in Greece. Since the literature strongly suggests a single-factor model of the CD-RISC-10, we performed confirmatory factor analysis to investigate the structure of the tool in our sample. Our CFA is consistent with the findings of other studies confirming the single-factor model of the tool. A great amount of literature supports our finding that the CD-RISC-10 is a unidimensional tool to measure resilience[51-57]. Our study investigated for first time the validity of the CD-RISC-10 in a sample of nurses. Previous studies investigated the validity of the tool in different study populations such as patients, healthy individuals, younger and older adults. Moreover, these studies conducted in Europe (United Kingdom, France, and Spain), North America (USA), Asia (China), and Africa (Nigeria). A similar study was performed recently in Greece but researchers assessed the validity of the CD-RISC-25 and not of CD-RISC-10 in health individuals and psychiatric patients[58].

Additionally, we found that the CD-RISC-10 has great internal consistency in our sample with Cronbach’s alpha and McDonald’s omega values greater than 0.92. Literature shows that reliability of CD-RISC-10 is excellent since Cronbach’s alpha ranged from 0.880 to 0.930 in studies in different countries[52-57]. Also, investigators that performed test-retest reliability analysis found very high intraclass correlation coefficients (greater than 0.800) supporting therefore the high reliability of CD-RISC-10.

Moreover, our multivariable analysis identified that levels of resilience was higher among nurses with higher educational level. In particular, nurses with a PhD/MSc diploma had higher score on CD-RISC-10 than those without a PhD/MSc diploma. This finding is consistent with a recent study where scholars found a positive correlation between resilience and educational level in a sample of intensive care units nurses across the United States [45]. In particular, 50% of registered nurses showed high levels of resilience, while the respective percentage for nursing assistants was 23%. Moreover, a study with 1012 nurses in Greece found that resilient nurses were better educated after elimination of several confounders[69]. Higher education offers nurses higher critical reflective practice and autonomy[70]. Additionally, highly educated nurses with a better theoretical knowledge can more easily develop and implement planful problem-solving and positive reappraisal in clinical setting[71]. In this context, high educational level can improve nurses’ resilience to deal with difficulties in clinical environments.

Furthermore, after eliminating confounders, we found that nurses with children presented with higher levels of resilience compared to those without children. This finding was also observed in the study by Afshari and colleagues[72]. One possible explanation for this finding could be that having children and worrying about the self and their children about getting infected with COVID-19 have been among the factors contributing to anxiety and stress in nurses, which may play an important role in reducing nurses’ resilience.

Concerning gender, in our sample, female nurses appeared more resilient than males, a relationship that was not statistically significant. Similar findings over the years have emphasized that despite the differences in scoring in the CD-RISC scales between males and females, gender does not affect one’s levels of resilience, as no statistically significant relations have been identified in corresponding studies (i.e., focusing on adult nurses, adult healthcare professionals) [39-44]. Interestingly, a study in healthcare professionals amid the challenges and adversities they faced when combating the COVID-19 pandemic concluded that neither gender nor age impact resilience capacity[73]. Our findings agree with that, as no statistically significant association was found between participants’ age and scoring on the scale.

References:
- Please include DOI for the references.

Answer: Done

It is crystal clear that this work needs much work in terms of literature review and interpretation of psychometric analysis. The number of references is very low. Please try to read and integrate more works.

Answer: Done

We added about 17 references by conducting a literature review and interpret the psychometric properties of the CD-RISK-10. Please, see the answers above and the manuscript (Introduction and Discussion section).

Author Response

Dear Reviewer,

Thank you very much for the peer review of the manuscript “Psychometric properties of the Greek Version of the Connor- Davidson Resilience Scale (CD-RISC-10) in a sample of nurses”. Thank you for your comments, which have improved the quality of the manuscript.

We have addressed all the comments (highlighted in yellow) in the revised text. Also, we made changes in the manuscript according to the other Reviewers’ instructions.

Please, find below an item-by-item answer to your comments. Hoping the revised manuscript fulfils the journal’s standards, we thank you for your courtesy.

We are looking forward to your response.

Best Regards

The authors

I want to thank you for the opportunity to review this manuscript. The time spent creating and submitting it is greatly appreciated. However, I consider that, despite the fact that the topic is interesting, the following changes are necessary for its possible publication:

Introduction: Please, describe studies that analyze the psychometric properties of the CD-RISC-10; Please, explain better all aims and hypotheses at the end of the introduction.

Answer: Done

We expanded the Introduction section providing studies that have examined the psychometric properties of CD-RISC-10 in different settings and populations. Moreover, we added the article that you mentioned explaining the reasons that a new translation was needed. Also, we presented our hypotheses in detail.

We add the following text in the introduction section:

In short, CD-RISC-25 has been widely used in a great amount of studies presenting good psychometric properties in different setting and countries. However, there is a significant disagreement regarding the factor structure of CD-RISC-25. For instance, Wu et al. found a 4-factor structure of the tool in a sample of employees[47], while Yu & Zhang found a 3-factor structure[48]. Since the factor structure of CD-RISC-25 has been unstable, Campbell-Sills & Stein refined CD-RISC-25 and established a shorter form, i.e. the CD-RISC-10 with ten items[49].

Afterwards, several studies examined the psychometric properties of CD-RISC-10 in different populations, settings, and cultures. In particular, Tu et al. confirmed the single-factor model of the CD-RISC-10 in a Chinese military personnel sample[50]. Also, they found that the CD-RISC-10 had a good concurrent and predictive validity using an anxiety and a depression scale. Several other studies confirmed the unidimensional factor structure of the CD-RISC-10[51-57]. Furthermore, the single-factor model of the CD-RISC-10 is strongly recommended since it is found in studies in different countries (e.g. USA, United Kingdom, China, France, Spain, Iran, Nigeria) with different study populations (e.g. healthy individuals, chronic patients, young adults, older adults). Additionally, a considerable amount of studies suggested that CD-RISC-10 has good convergent, criterion and divergent validity using a variety of tools measuring social support, self-efficacy, self-esteem, mastery, mental health, depression, and anxiety[50-51,53-57]. For instance, Neyer et al. found a strong positive correlation (r=0.78) between CD-RISC-10 and another tool that measures resilience, namely 14-item Resilience Scale[55]. Moreover, several studies showed the excellent internal consistency and reliability of CD-RISC-10 by calculating Cronbach’s alpha coefficient and performing test-retest reliability analysis. In particular, Cronbach’s alpha coefficient ranged from 0.880 to 0.93[52-57]. Also, intraclass correlation coefficients in test-retest reliability analysis ranged from 0.800 to 0.900. In conclusion, CD-RISC-10 has superior psychometric properties [51-57]and some scholars suggested the replacement of CD-RISC-25 with the CD-RISC-10 to measure resilience in studies[49,51,53].

A recent study examined the psychometric properties of Greek version of CD-RISC-25 in a sample of 244 healthy participants and 303 psychiatric patients[58]. However, until now the Greek version of CD-RISC-10 has not been validated. Moreover, literature suggests that resilience among nurses is a significant internal psychological factor that promotes their ability to cope with difficult situations[59-62]. Thus, validation of CD-RISC-10 in a sample of nurses can give scholars a brief, simple, and valid tool to measure nurses’ resilience. Therefore, we aimed to translate into Greek and validate the psychometric properties of the CD-RISC-10 and investigate the resilience levels among a convenience sample of nurses currently working in the Greek public healthcare system. As mentioned above, literature suggests a single-factor model of the CD-RISC-10. In this context, we expected also a single-factor model of the Greek version of CD-RISC-10. For that reason, we performed confirmatory factor analysis to establish the structure of the Greek version of CD-RISC10. Moreover, we hypothesized that different levels of resilience could be arisen according to several demographic and job characteristics, namely gender, age, living arrangement, children, job position, educational level, and clinical experience. Finally, we examined the measurement invariance of CD-RISC-10 since evidence on this issue is limited.

Method: Please, describe the screening of data (skewness and kurtosis)

Answer: Done

We added skewness and kurtosis values in Table 2. Moreover, we added the following text in Results section:

Skewness values for the ten items ranged from -0.44 to 0.06, while kurtosis values ranged from -1.00 to 0.07. Therefore, the ten items followed normal distribution since skewness and kurtosis values were between -1.00 and 1.00.

It would be useful to compare your factor structure with other models in the literature. In the discussion section, a comparison of its review with the initial contextualization is not made. Personally, I am not in favor of using literature in the discussion that has not been previously mentioned in the introduction, since it conveys an incomplete review of the state of the investigated topic.

Answer: Done

We rewrote the discussion according to your comment. We compared our factor structure with other models in the literature. Moreover, we expanded the Introduction section providing studies that have examined the psychometric properties of CD-RISC-10 in different settings and populations. Then, we used the studies in the Introduction section to discuss our results in the Discussion section.

We rewrite the Discussion section as following.

Our aim was to translate and validate the CD-RISC-10 in a sample of nurses in Greece. Since the literature strongly suggests a single-factor model of the CD-RISC-10, we performed confirmatory factor analysis to investigate the structure of the tool in our sample. Our CFA is consistent with the findings of other studies confirming the single-factor model of the tool. A great amount of literature supports our finding that the CD-RISC-10 is a unidimensional tool to measure resilience[51-57]. Our study investigated for first time the validity of the CD-RISC-10 in a sample of nurses. Previous studies investigated the validity of the tool in different study populations such as patients, healthy individuals, younger and older adults. Moreover, these studies conducted in Europe (United Kingdom, France, and Spain), North America (USA), Asia (China), and Africa (Nigeria). A similar study was performed recently in Greece but researchers assessed the validity of the CD-RISC-25 and not of CD-RISC-10 in health individuals and psychiatric patients[58].

Additionally, we found that the CD-RISC-10 has great internal consistency in our sample with Cronbach’s alpha and McDonald’s omega values greater than 0.92. Literature shows that reliability of CD-RISC-10 is excellent since Cronbach’s alpha ranged from 0.880 to 0.930 in studies in different countries[52-57]. Also, investigators that performed test-retest reliability analysis found very high intraclass correlation coefficients (greater than 0.800) supporting therefore the high reliability of CD-RISC-10.

Moreover, our multivariable analysis identified that levels of resilience was higher among nurses with higher educational level. In particular, nurses with a PhD/MSc diploma had higher score on CD-RISC-10 than those without a PhD/MSc diploma. This finding is consistent with a recent study where scholars found a positive correlation between resilience and educational level in a sample of intensive care units nurses across the United States [45]. In particular, 50% of registered nurses showed high levels of resilience, while the respective percentage for nursing assistants was 23%. Moreover, a study with 1012 nurses in Greece found that resilient nurses were better educated after elimination of several confounders[69]. Higher education offers nurses higher critical reflective practice and autonomy[70]. Additionally, highly educated nurses with a better theoretical knowledge can more easily develop and implement planful problem-solving and positive reappraisal in clinical setting[71]. In this context, high educational level can improve nurses’ resilience to deal with difficulties in clinical environments.

Furthermore, after eliminating confounders, we found that nurses with children presented with higher levels of resilience compared to those without children. This finding was also observed in the study by Afshari and colleagues[72]. One possible explanation for this finding could be that having children and worrying about the self and their children about getting infected with COVID-19 have been among the factors contributing to anxiety and stress in nurses, which may play an important role in reducing nurses’ resilience.

Concerning gender, in our sample, female nurses appeared more resilient than males, a relationship that was not statistically significant. Similar findings over the years have emphasized that despite the differences in scoring in the CD-RISC scales between males and females, gender does not affect one’s levels of resilience, as no statistically significant relations have been identified in corresponding studies (i.e., focusing on adult nurses, adult healthcare professionals) [39-44]. Interestingly, a study in healthcare professionals amid the challenges and adversities they faced when combating the COVID-19 pandemic concluded that neither gender nor age impact resilience capacity[73]. Our findings agree with that, as no statistically significant association was found between participants’ age and scoring on the scale.

Round 2

Reviewer 2 Report

The corrections are thorough; thank you for your work and congratulations on the paper.

Reviewer 3 Report

Accept in present form